# Neurodevelopment in children born to women exposed to pesticides during pregnancy

William Nelson Mwakalasya[1], Simon Henry Mamuya[1], Karim Manji[2],
Bente Elisabeth Moen[3]*, Aiwerasia Vera Ngowi[1]

1 Department of Environmental and Occupational Health, Muhimbili University of Health and Allied Sciences, Dar es salaam, Tanzania, 2 Department of Pediatrics and Child Health, Muhimbili University of Health and Allied Sciences, Dar es Salaam, Tanzania, 3 Department of Global Public Health and Primary Care, Centre for International Health, University of Bergen, Bergen, Norway

* bente.moen@uib.no

## Abstract

The global rise in pesticide use, particularly across Africa, raises concerns about maternal occupational exposure during pregnancy and its potential impact on child neurodevelopment. This study examined associations between self-reported maternal pesticide exposure during pregnancy and neurodevelopmental outcomes in children aged 4–6 years. A cross-sectional design was implemented, comprising 432 mother–child pairs from three horticulture-intensive regions in Tanzania. Maternal exposure was assessed through structured interviews, focusing on activities during pregnancy such as pesticide spraying, weeding, and handling contaminated clothing. Children's neurodevelopment was evaluated using the International Development and Early Learning Assessment (IDELA), which measures motor skills, literacy, numeracy, social–emotional development, and executive function. Linear regression analyses revealed that children of mothers who reported pesticide spraying during pregnancy scored significantly lower in social–emotional ($\beta = -6.813$, 95% CI: $-11.53$ to $-2.096$, $p = 0.005$) and executive function ($\beta = -9.317$, 95% CI: $-16.007$ to $-2.627$, $p = 0.006$) domains. Overall, children achieved the highest mean scores in fine and gross motor skills ($62.11 \pm 19.3$) and the lowest in executive function ($43.97 \pm 24.3$). Age-related differences were also observed, with six-year-olds consistently outperforming younger children across all developmental domains. These findings suggest an association between maternal pesticide exposure during pregnancy and adverse neurodevelopmental outcomes in offspring. Given that exposure data were self-reported, results should be interpreted cautiously. Nevertheless, the study underscores the urgent need for comprehensive risk assessments incorporating objective exposure measurements, particularly in horticultural settings where women of reproductive age represent a substantial proportion of the workforce.

**Data availability statement:** Our data involve sensitive human research participant data for mother and child, and might be identifying. Therefore the data cannot be publicly shared, as there are restrictions in accordance with the ethical committees in Tanzania and Norway. However, the data set can be made available upon request, by contacting The Regional Committee for Medical and Health Research Ethics in Norway - REK-South East, P.Box 1130, Blindern, 0318 Oslo, Norway, ass. director e-mail address j.c.holen@medisin.uio.no.

**Funding:** This research was funded by the Norwegian Programme for Capacity Development in Higher Education and Research for Development (Norad). NORHED II – SAFEWORKERS Project, grant number: 69181. The funders had no role in study design, data collection and analysis, decision to publish, or preparation of the manuscript.

**Competing interests:** The authors have declared that no competing interests exist.

## Introduction

Global pesticide use has risen by about 80% in the last two decades [1], with Africa experiencing a staggering 175% increase, largely due to the growing population that put pressure on the agriculture sector to ensure sufficient yields in crop production [2]. However, pesticide use in Africa is associated with severe health consequences for agricultural workers and their families, including increased healthcare costs and long-term adverse effects [3,4]. Despite these risks, most African countries lack the resources to effectively monitor or mitigate the dangers of pesticide exposure, leaving vulnerable populations at risk.

Women are integral to agriculture, especially in low- and middle-income countries. In sub-Saharan Africa, they make up approximately 55% of the agricultural workforce, with Tanzania having the highest proportion at 81% [5]. Within the horticultural sector, women face an increased risk of pesticide exposure through activities such as planting, weeding, harvesting, and pesticide application [6]. These activities vary due to different needs in the farm. The women may have specific work tasks like weeding but can be asked to perform other tasks for shorter or longer periods. Compared to men, women are often more vulnerable to pesticide exposure because they receive less training, and their work is generally perceived as less hazardous, resulting in inadequate protection measures [7]. Exposure pathways include direct contact with pesticides during pesticide mixing and application, as well as indirect exposure from residues on crops, soil and contaminated environments. Literature shows that pesticides exposure can cause severe acute and chronic health effects in humans, including neurotoxicity, carcinogenicity, endocrine disruption, and respiratory issues, with agricultural workers and communities in developing countries facing disproportionate exposure [8,9].

Horticultural work during pregnancy is common in many settings [10,11], and such activities may expose pregnant women to pesticides. Of particular concern is the potential for several of these pesticides to cross the placental barrier, leading to fetal exposure and adverse pregnancy or developmental outcomes [12,13]. The fetus is especially vulnerable to pesticide toxicity due to underdeveloped detoxification mechanisms; for instance, paraoxonase 1 (PON1), an enzyme critical for metabolizing organophosphate pesticides, does not reach adult levels until approximately two years of age [14]. In Tanzania, several pesticides with documented reproductive, developmental, and neurotoxic effects were reported to be in use in 2020, for instance 2,4-D amine salts, chlorpyrifos, pirimicarb, permethrin, and thiacloprid [15]. There is a documented evidence of the use of these pesticides in horticulture in Tanzania, where over 80% of the pesticides reported to be used in northern Tanzania contained compounds classified as moderately to highly hazardous [16]. Studies have shown relationship between prenatal exposure to hazardous pesticides and neurodevelopmental impairments in children [17,18], mediated through mechanisms such as acetylcholinesterase (AChE) inhibition [19]. Prenatal exposures are also linked with damage of the nervous system through several other non-cholinesterase mechanisms. These include impairing axonal transport, inducing oxidative stress, causing mitochondrial dysfunction, and triggering neuroinflammation [20–22].

However, some studies report no significant association between pesticide exposure and neurodevelopment, including research on organophosphate-exposed 2-year-old Chinese children and chlorpyrifos-exposed 36-month-old children of African American and Dominican women in the USA [23,24].

Given these inconsistencies in the literature, further investigation is warranted to clarify the potential risks of prenatal pesticide exposure on neurodevelopmental outcomes. Pesticide exposure assessment studies vary, as some focus on exposure during the first and second trimesters, while others assess exposure during later stages of pregnancy. Also, neurodevelopmental outcomes have been evaluated by diverse assessment tools across different age groups, but standardized tools have rarely been used [25] Also, children aged 4–6 years have rarely been studied, particularly not in sub-Saharan Africa. This study aims to investigate the relationship between self-reported maternal pesticide exposure during pregnancy and neurodevelopmental outcomes, assessed using the International Development and Early Learning Assessment (IDELA), in their children aged 4–6 years in Tanzania. The use of self-reported exposure is particularly relevant for designing practical risk communication and mitigation strategies in resource-limited settings where biomonitoring is often not feasible.

## Materials and methods

### Study design and population

This was a cross-sectional analytical study conducted in three regions (A, B and C) of Tanzania. The study was conducted among children born of women working in small-scale horticulture farms who had been working with pesticides before and during conception. Study participants were recruited from two horticulture-intensive wards per district. This is a part of a larger study, and a detailed study plan is described elsewhere [10]. A systematic sampling technique was employed whereby every third household was selected and included in the study. A total of 432 mother-child pairs were obtained, distributed proportionally in the districts based on horticultural land size. Data was collected by five trained research assistants using a digital questionnaire on tablets, with data uploaded daily to Kobo Toolbox servers. Information about the children was collected from their mothers but in order to obtain children's nutrition status, weight and height of a child was measured. The data collection started 1st November 2022 and ended 15th April 2023.

### Maternal pesticides exposure assessment

Self-reported pesticides exposure during pregnancy was done using a semi-structured closed-ended questionnaire. The questionnaire was used in order to gain information about work and potential exposure to pesticides, as this information is not routinely collected in the farm or health facility records. The questionnaire was divided into two sections. The first section gathered background details such as age, alcohol consumption, cigarette smoking, proximity to the farm, years of experience in horticulture, years lived in the horticulture community, pregnancy-related hospital visits and delivery complications. Delivery complications included conditions such as obstructed labor, severe bleeding and sepsis, as reported by the mother. The second section examined activities performed by women that could contribute to pesticide exposure during pregnancy. Participants were asked to recall whether, during the pregnancy of the child under study, they engaged in any of the following activities: mixing or diluting pesticides, spraying pesticides, weeding or harvesting within 24 hours after pesticide application, washing spray equipment, laundering pesticide-contaminated clothing, or consuming vegetables or fruits within 24 hours of pesticide application.

### Neurodevelopmental assessment

The International Development and Early Learning Assessment (IDELA), developed by Save the Children, was used to measure children's learning and development across various domains, including motor, literacy, numeracy, and social-emotional development (Table 1). IDELA is a child centered tool that comprises 24 core items designed to directly

**Table 1. Components of the IDELA domains.**

| Domain | Items | Total possible points |
|---|---|---|
| Social and emotional development | Self-awareness | 6 |
| | Friends | 10 |
| | Emotional awareness/regulation | 5 |
| | Empathy/perspective taking | 3 |
| | Solving conflict | 2 |
| Emergent numeracy | Comparison by size and length | 4 |
| | Sorting and classification | 2 |
| | Shape identification | 5 |
| | Number identification | 20 |
| | One-to-one correspondence | 5 |
| | Addition and subtraction | 3 |
| | Puzzle completion | 8 |
| Emergent literacy | Expressive vocabulary | 20 |
| | Print awareness | 3 |
| | Letter identification | 20 |
| | First letter sounds | 3 |
| | Emergent writing | 4 |
| | Oral comprehension | 7 |
| Fine and Gross Motor function | Copying a shape | 4 |
| | Drawing a person | 10 |
| | Folding paper | 6 |
| | Hopping | 10 |
| Executive function | Short-term memory | 4 |
| | Inhibitory control | 13 |

assess key developmental skills and early learning in preschool children aged 3.5 to 6.5 years. For each item, the percentage of the correct scores was calculated for each task/activity by taking the number of correct responses and dividing by the total possible. Some tasks were dichotomous (score 0 or 1), whereas other tasks have different number of points possible. The average percentage for each domain was calculated by simply dividing the sum the percentage for each of the tasks in the domain by the number of tasks. The overall IDELA score was obtained by taking the average of all of the five domain scores. The IDELA tool has already been utilized in over 70 countries including Tanzania. [26] demonstrating its adaptability to diverse cultural contexts and its strong reliability and validity. For this study, the original English version of IDELA was translated into Kiswahili, the native language of both the researchers and the study participants. The assessment was conducted at home by trained research assistants, who first obtained the child's assent before securing verbal consent from the mother or caregiver. The mother/caregiver remained present throughout the assessment to ensure the child felt safe and comfortable.

## Covariates

Covariates were selected to be considered in the analysis based on their association with neurodevelopment scores in this study. Since only child's age (a continuous variable in years) showed a significant association with neurodevelopmental domains in bivariate analysis, it is the only covariate included in the analysis. The rationale for inclusion was their potential confounding influence on both pesticide exposure patterns and child neurodevelopmental outcomes.

## Statistical analysis

Descriptive statistics were used to summarize the independent variables, which included sociodemographic factors and pesticide exposure activities reported by women during pregnancy. The dependent variable was the IDELA scores, representing the points accumulated by children across five neurodevelopmental domains: social-emotional, emergent numeracy, emergent literacy, fine and gross motor skills, and executive function. These scores were converted into percentages, and total scores were calculated by summing points from all domains. Higher domain mean scores indicated better performance.

The Shapiro-Wilk test for normality revealed that the neurodevelopmental scores were not normally distributed ($p < 0.05$). Consequently, a nonparametric Independent-Samples Mann-Whitney U Test was used for a univariate analysis to examine the relationship between the independent variables, and the dependent variable mean scores. The significance level was set to $p < 0.01$, due to the large number of analyses. Additionally, a multiple linear regression analysis was conducted to assess the relationship between the independent and dependent variables (domain specific and total neurodevelopment score). Independent variables were the pesticide exposure-related activities and maternal and child social demographic characteristics. The dependent variable was each of the neurodevelopment scores; they were analyzed separately. Despite the non-normal distribution of the raw scores, linear regression was employed due to the central limit theorem's applicability with large sample sizes ($n = 432$) and its advantage in producing easily interpretable coefficients. Diagnostic plots (Q-Q plots, scatterplots of residuals vs. predicted values) were examined for the final models and indicated that the residuals were approximately normally distributed and homoscedastic. All statistical analyses were performed using the Statistical Package for the Social Sciences (SPSS), version 27.

## Ethics approval and consent to participate

The study was approved by the Institutional Review Board of Muhimbili University of Health and Allied Sciences (MUHAS) (MUHAS-REC-08-2022-1332) and the Regional Committee for Research Ethics of Southeast Norway (535644/2023). District executive directors granted permission to access wards, villages, and households, issuing letters to inform Ward Executive Officers (WEOs), who in turn notified hamlet leaders by phone. During household or farm visits, research assistants explained the study's purpose, and participants provided verbal consent before the interview. The consent was documented in written by a researcher, and witnessed by a supervisor with writing skills at the workplace. During neurodevelopmental assessment a child had to assent before obtaining a verbal consent from a mother/caregiver.

## Results

### Social-demographic characteristics of study participants

In this study, a total of 432 mother-child pairs out of 436 reached (99% response rate) were included in the analysis. Four women declined to participate, citing dissatisfaction with the perceived benefits of the study. Most mothers were aged 31–40 years (55.1%) and over half (55.3%) had six to ten years of experience in horticulture, but most (39.8%) have been living in the area for more than 20 years. Delivery complications were reported by only few (3.7%) women. Children were dominated by those aged 6 years (43.5%) and sex distribution was almost equal with girls slightly outnumbering boys (53% vs. 47%). Re-entry in the farm for weeding within 24 hours after spray (57.2%) and washing clothes used for pesticide spray (51.6%) are the activities most reported by women (**Table 2**).

### Children's neurodevelopment performance

The mean IDELA scores by domain ranged between 43.59 to 62.11 with an overall mean of 50.69. Fine and gross motor domain had the highest mean score of 62.11 ± 19.3 followed by emergent numeracy with a mean score of 57.04 ± 23.1 and the lowest mean scores was achieved in emergency literacy (43.59 ± 20.7) and executive function (43.97 ± 24.3) domains (Fig 1).

**Table 2. Social demographic characteristics of 432 mother and child pairs.**

| Variables | Categories | Frequency (%) |
|---|---|---|
| **Maternal characteristics** | | |
| Study areas | A | 80 (18.5) |
| | B | 258 (59.7) |
| | C | 94 (21.8) |
| Mother's age (years) | Under 30 | 141 (32.6) |
| | 31 - 40 | 238 (55.1) |
| | Above 40 | 53 (12.3) |
| Years in horticulture | Under 5 | 122 (28.2) |
| | 6 to 10 | 239 (55.3) |
| | Above 10 | 71 (16.4) |
| Delivery complications | No | 416 (96.3) |
| | Yes | 16 (3.7) |
| Years lived in the area | Under 10 | 138 (31.9) |
| | 11 to 20 | 122 (28.2) |
| | Above 20 | 172 (39.8) |
| **Children characteristics** | | |
| Age (years) | 4 | 104 (24.1) |
| | 5 | 140 (32.4) |
| | 6 | 188 (43.5) |
| Sex | Boys | 203 (47.0) |
| | Girls | 229 (53.0) |
| Nutrition status | Normal ($-1 \geq WHZ \leq +1$) | 330 (77.1) |
| | Wasted ($WHZ < -1$) | 54 (12.6) |
| | Overweight ($WHZ > +1$) | 44 (10.3) |
| **Work with possible pesticides exposure practiced during pregnancy** | | |
| Mixing pesticides | No | 398 (92.1) |
| | Yes | 34 (7.9) |
| Spraying pesticides | No | 269 (62.3) |
| | Yes | 163 (37.7) |
| Weeding within 24 hours after spray | No | 185 (42.8) |
| | Yes | 247 (57.2) |
| Harvesting within 24 hours after spray | No | 335 (77.5) |
| | Yes | 97 (22.5) |
| Washed clothes used in pesticides work | No | 209 (48.4) |
| | Yes | 223 (51.6) |
| Washing spray equipment after use | No | 302 (69.9) |
| | Yes | 130 (30.1) |
| Eating crops within 24 hours after spray | No | 235 (54.4) |
| | Yes | 197 (45.6) |

Independent-Samples Mann-Whitney U Test showed that child's age was significantly associated with the neurodevelopment scores across all domains ($p \leq 0.01$) (Table 3). Older children (6 years) consistently exhibited higher mean scores across all domains compared to younger children, with six-year-olds displaying the highest scores. The higher scores observed in the 'wasted' children are likely attributable to their slightly older mean age (5.7 years) compared to the

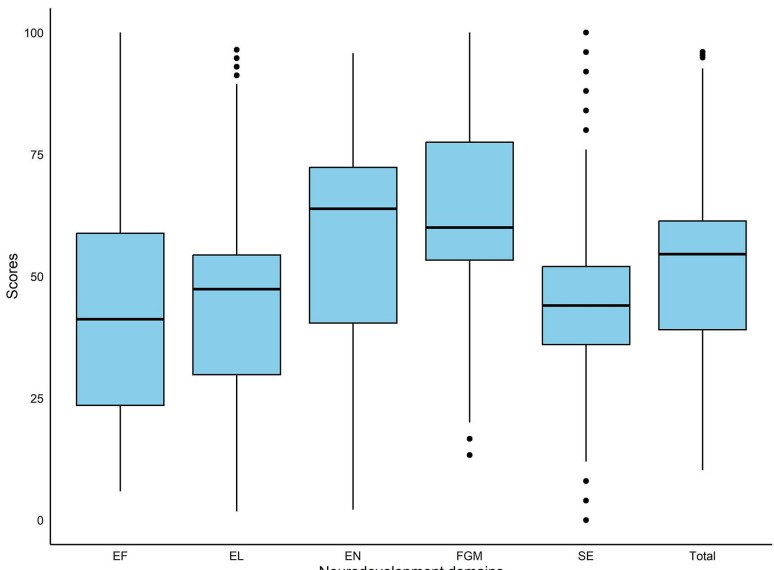

**Fig 1. A boxplot showing the distribution of neurodevelopment scores (%) for each domain. SE – Social emotional; EN – Emotional Numeracy; EL – Emergent Literacy; FGM – Fine and Gross Motor; EF – Executive Function.**

'normal' (5.4 years) and 'overweight' (5.3 years) children. Other sociodemographic factors, such as study area, mother's age, child's sex, years in horticulture, years lived in the area, and delivery complications, did not show statistically significant differences with neurodevelopment scores.

### The association between maternal pesticide exposure and the offspring's neurodevelopment (performance)

Women who reported to engage in pesticides spray activity during pregnancy had children with significantly lower mean emergent numeracy (53.88 ± 20.1) and lower fine and gross motor domain scores (58.77 ± 17.8) compared to those who reported to not spray during pregnancy (Mann-Whitney U-test, p ≤ 0.01) (Table 4). Similar outcomes were observed in children of women who reported washing clothes used for pesticides spray/handling activities. Mean emergent numeracy domain scores were also significantly lower (54.88 ± 22.6) in children of mothers who reported to engage in weeding activities within 24 hours after pesticide spray during pregnancy, compared to those who reported not to (59.93 ± 23.4). Other horticulture practices like mixing pesticides, washing equipment used for pesticides work and eating crops sprayed with pesticides within 24 hours did not show any statistically significant relationship with the neurodevelopment scores.

A multiple linear regression, with all independent variables included in the model simultaneously, was run to predict neurodevelopment scores from reported pesticide exposure activities using the child's age as a covariate (Fig 2). The analysis showed a borderline relationship between engaging in spray work during pregnancy and low total test scores (β = −4.127, 95% CI: −8.256 to 0.002, p = 0.05), while no other statistical relationships were found. It was also shown that spraying pesticides during pregnancy was significantly associated with lower social-emotional scores (β = −6.813, 95% CI: −11.53 to −2.096, p = 0.005) and executive function scores (β = −9.317, 95% CI: −16.007 to −2.627, p = 0.006). On the other hand, children of women who reported doing weeding within 24 hours after spraying showed a significant positive association with social-emotional scores (β = 4.774, 95% CI: 0.174 to 9.373, p = 0.042). Emergency numeracy, emergency literacy and fine and gross motor scores were not associated to any of the variables in the model. Data showing the unadjusted and adjusted linear regression models are summarized in S1 Table.

**Table 3.  The association between maternal and child sociodemographic variables and mean neurodevelopment scores in 432 children aged 4–6 years.**

| Sociodemographic variables | Categories | SE | EN | EL | FGM | EF | Total |
|---|---|---|---|---|---|---|---|
| Study areas | A (n = 258) | 44.54 ± 15.3 | 57.41 ± 22.6 | 44.52 ± 18.9 | 62.47 ± 18.6 | 41.09 ± 21.5 | 50.70 ± 16.9 |
| | B (n = 94) | 46.89 ± 20.8 | 55.61 ± 23.8 | 42.40 ± 24.2 | 62.17 ± 20.1 | 47.62 ± 26.9 | 50.44 ± 20.4 |
| | C (n = 80) | 48.55 ± 19.8 | 57.52 ± 24.0 | 41.97 ± 21.9 | 60.92 ± 20.7 | 48.97 ± 28.0 | 50.97 ± 18.8 |
| Mother's age (years) | Under 30 (n = 141) | 45.93 ± 17.6 | 59.54 ± 24.3 | 44.00 ± 21.7 | 60.92 ± 20.7 | 46.10 ± 25.7 | 51.51 ± 19.3 |
| | 31–40 (n = 238) | 46.05 ± 17.8 | 56.09 ± 22.5 | 43.93 ± 20.4 | 62.49 ± 18.8 | 43.18 ± 23.9 | 50.57 ± 17.6 |
| | Above 40 (n = 53) | 44.30 ± 16.5 | 54.68 ± 22.5 | 41.01 ± 19.7 | 63.59 ± 19.1 | 41.84 ± 22.0 | 49.06 ± 16.7 |
| Child's age (years) | 4 (n = 104) | 30.93 ± 15.8* | 31.51 ± 21.6* | 21.59 ± 15.8* | 44.41 ± 17.8* | 26.72 ± 19.9* | 29.95 ± 14.6* |
| | 5 (n = 140) | 46.29 ± 15.0* | 59.69 ± 18.2* | 44.28 ± 17.0* | 63.21 ± 15.7* | 41.80 ± 23.5* | 51.67 ± 13.5* |
| | 6 (n = 188) | 53.93 ± 14.6* | 69.64 ± 13.7* | 55.38 ± 14.8* | 71.32 ± 15.3* | 55.24 ± 20.9* | 61.69 ± 11.3* |
| Child's sex | Boy (n = 203) | 46.05 ± 16.9 | 57.74 ± 22.8 | 44.23 ± 21.3 | 62.92 ± 19.7 | 44.16 ± 24.5 | 51.28 ± 17.9 |
| | Girl (n = 229) | 45.57 ± 18.1 | 56.42 ± 23.4 | 43.02 ± 20.2 | 61.40 ± 18.9 | 43.80 ± 24.1 | 50.17 ± 18.2 |
| Child's nutrition status | Normal (n = 330) | 44.58 ± 15.4 | 57.00 ± 22.1 | 43.60 ± 18.7 | 60.76 ± 18.5 | 41.68 ± 22.5* | 50.06 ± 16.7 |
| | Wasted (n = 54) | 53.11 ± 22.6 | 61.35 ± 24.6 | 44.15 ± 27.2 | 67.72 ± 21.2 | 56.43 ± 28.9* | 55.22 ± 20.5 |
| | Overweight (n = 44) | 46.00 ± 23.5 | 51.40 ± 28.2 | 41.95 ± 26.0 | 65.45 ± 21.7 | 45.45 ± 27.4* | 49.39 ± 23.8 |
| Years in horticulture | Under 5 (n = 122) | 45.77 ± 16.7 | 59.85 ± 21.6 | 44.00 ± 19.3 | 63.80 ± 17.4 | 41.90 ± 22.2 | 51.66 ± 16.3 |
| | 6 to 10 (n = 239) | 45.79 ± 18.2 | 54.40 ± 24.0 | 42.18 ± 21.4 | 60.36 ± 20.4 | 44.11 ± 25.7 | 49.24 ± 19.0 |
| | Above 10 (n = 71) | 45.86 ± 17.0 | 61.10 ± 21.7 | 47.64 ± 20.3 | 65.12 ± 18.1 | 47.06 ± 22.5 | 53.91 ± 17.1 |
| Years lived in the area | Under 10 (n = 138) | 47.22 ± 20.3 | 56.00 ± 23.2 | 41.69 ± 23.4 | 60.65 ± 20.6 | 47.36 ± 27.8 | 50.07 ± 19.7 |
| | 11 to 20 (n = 122) | 46.13 ± 15.2 | 59.47 ± 22.5 | 45.07 ± 17.5 | 61.69 ± 18.1 | 42.43 ± 22.3 | 51.64 ± 15.9 |
| | Above 20 (n = 172) | 44.42 ± 16.7 | 56.16 ± 23.5 | 44.07 ± 20.5 | 63.59 ± 19.1 | 42.34 ± 22.4 | 50.51 ± 18.1 |
| Delivery complications | No (n = 416) | 45.74 ± 17.6 | 57.00 ± 23.2 | 43.33 ± 20.5 | 61.98 ± 19.3 | 43.91 ± 24.3 | 50.56 ± 18.0 |
| | Yes (n = 16) | 47.25 ± 17.0 | 58.11 ± 21.1 | 50.44 ± 25.1 | 65.63 ± 19.8 | 45.59 ± 25.2 | 54.15 ± 19.0 |

*p < 0.01 in Independent-Samples Mann-Whitney U Test. SE – Social emotional; EN – Emotional Numeracy; EL – Emergent Literacy; FGM – Fine and Gross Motor; EF – Executive Function

Sensitivity analysis was conducted to check whether the observed neurodevelopmental effects were sex-specific. The analysis revealed that the significant negative association between spraying pesticides and the observed decrease in social-emotional and executive function scores is driven primarily by boys (S2 Table). Boys of mothers who reported spraying pesticides during pregnancy had significantly lower social emotional (β = −8.227, 95%CI: −14.267 to −2.188, p = 0.008), executive function (β = −11.705, 95%CI: −10.082 to −2.328, p = 0.015) and total IDELA scores (β = −5.868, 95%CI: −11.544 to −0.192, p = 0.043). None of the pesticide exposure activities showed statistically significant associations with neurodevelopmental scores in girls, though the point estimates for several activities were negative.

## Discussion

Our results show that self-reported maternal exposure to pesticides through direct spraying during pregnancy was associated with lower scores in social-emotional and executive function domains among children aged 4 to 6 years. We also found an association between the social-emotional scores among the children and of the engagement in weeding among their mothers during pregnancy. Additionally, direct spraying of pesticides showed a borderline association with reduced overall neurodevelopmental scores.

These results are consistent with a previous study conducted among children of smallholder tomato farmers in southern Tanzania, which reported delayed neurodevelopment in children associated with maternal engagement in agricultural work [27]. However, our present study provides more granular insight by identifying direct pesticide spraying as a

**Table 4. The association between maternal pesticide exposure variables and mean neurodevelopment scores in children aged 4–6 years.**

| Pesticides exposure activities practiced during pregnancy | Categories | SE | EN | EL | FGM | EF | Total |
|---|---|---|---|---|---|---|---|
| Mixing pesticides | No (n=398) | 45.82±17.7 | 57.30±23.4 | 43.45±20.7 | 62.55±19.3 | 43.91±24.2 | 50.79±18.2 |
| | Yes (n=34) | 45.53±15.3 | 54.01±19.3 | 45.20±20.2 | 57.06±18.3 | 44.64±25.4 | 49.57±15.6 |
| Spraying pesticides | No (n=269) | 46.30±18.8 | 58.96±24.6* | 44.03±21.6 | 64.14±19.9* | 44.54±25.4 | 51.82±19.4 |
| | Yes (n=163) | 44.96±15.3 | 53.88±20.1* | 42.87±19.2 | 58.77±17.8* | 43.02±22.4 | 48.83±15.4 |
| Weeding within 24 hours after spray | No (n=185) | 45.02±16.2 | 59.93±23.4* | 45.04±19.5 | 64.76±19.0 | 42.58±22.8 | 52.13±18.1 |
| | Yes (n=247) | 46.38±18.5 | 54.88±22.6* | 42.51±21.5 | 60.13±19.3 | 45.01±25.3 | 49.61±18.0 |
| Harvesting within 24 hours after spray | No (n=335) | 45.98±17.1 | 57.65±23.1 | 44.18±20.4 | 63.00±19.3 | 43.76±23.7 | 51.20±17.8 |
| | Yes (n=97) | 45.15±19.0 | 54.95±23.2 | 41.54±21.5 | 59.07±19.1 | 44.69±26.3 | 48.93±18.6 |
| Washed clothes used in pesticides work | No (n=209) | 44.86±16.8 | 59.03±24.1* | 45.03±20.1 | 64.32±19.2* | 42.08±22.8 | 51.75±18.2 |
| | Yes (n=223) | 46.67±18.2 | 55.18±22.0* | 42.25±21.2 | 60.04±19.2* | 45.74±25.4 | 49.70±17.8 |
| Washing spray equipment after use | No (n=302) | 45.14±16.7 | 57.46±23.1 | 44.33±20.0 | 62.41±19.4 | 42.46±22.6 | 50.85±17.7 |
| | Yes (n=130) | 47.32±19.3 | 56.07±23.0 | 41.88±22.1 | 61.44±19.1 | 47.47±27.5 | 50.31±18.7 |
| Eating crops within 24 hours after spray | No (n=235) | 45.09±15.5 | 58.76±22.0 | 44.68±19.1 | 63.13±18.9 | 42.53±22.0 | 51.44±16.8 |
| | Yes (n=197) | 46.64±19.7 | 55.00±24.3 | 42.29±22.4 | 60.90±19.8 | 45.69±26.6 | 49.80±19.4 |

*p<0.01 in Independent-Samples Mann-Whitney U Test. SE – Social and emotional development; EN – Emergent Numeracy; EL – Emergent Literacy; FGM – Fine and Gross Motor function; EF – Executive Function

significant contributor to maternal exposure. This distinction helps bridge an important gap in understanding how specific horticultural practices during pregnancy relate to adverse neurodevelopmental outcomes in children.

Adverse effects of pesticide exposure on social-emotional development have also been shown in several other studies. In Costa Rica, a prospective neurodevelopmental assessment of 355 one-year-old infants, born to mothers living within 5 km of banana plantations with frequent aerial mancozeb (a fungicide) spraying, found that higher prenatal urinary ethylenethiourea (a mancozeb metabolite) concentrations were associated with lower social-emotional scores, particularly among girls [28]. Similarly, a study analyzing 618 urine samples from women exposed to insecticides used in indoor residual spraying for malaria control in Limpopo, South Africa, revealed that maternal urinary metabolites of certain pyrethroids, specifically cis–(2,2-dichlorovinyl)-2,2 dimethylcyclopropane-1-carboxylicacid (DCCA), *trans*-DCCA, and 3-phenoxybenzoic acid (3PBA), were associated with lower social-emotional scores in one-year-old infants [29]. Postnatal exposure to organochlorine pesticides through breastfeeding has also been linked to impaired social-emotional development [30], though prenatal exposure did not show the same association [29]. Supporting this, a study of 55 Taiwanese infants aged 8–12 months found that higher breast milk levels of some organochlorines were significantly associated with poorer social-emotional performance [30]. Though this study did not perform biological monitoring, there is evidence that horticulture farmers in Tanzania use neurotoxic pesticides such as organophosphates; for instance chlorpyrifos and profenofos, carbamates; for instance carbaryl, pyrethroids; for instance lambda-cyhalothrin, and dithiocarbamates; for instance mancozeb [16], suggesting that both prenatal and postnatal pesticide exposure may adversely affect early social-emotional development.

In accordance with the present results on the association between pesticides exposure and executive functioning in children, previous studies have demonstrated similar results. In a U.S birth cohort involving 162 mother-child pairs found that prenatal exposure to pyrethroids, as indicated by its urinary metabolites 3-PBA and cis-DCCA, was associated with poorer performance in executive function in children aged 4–9 years [31]. Similarly, the CHAMACOS cohort study involving 363 mother-child pairs reported that higher gestational concentrations of organophosphate metabolites, as measured by maternal urine levels of dialkyl phosphate, were associated with significantly lower executive functioning in children

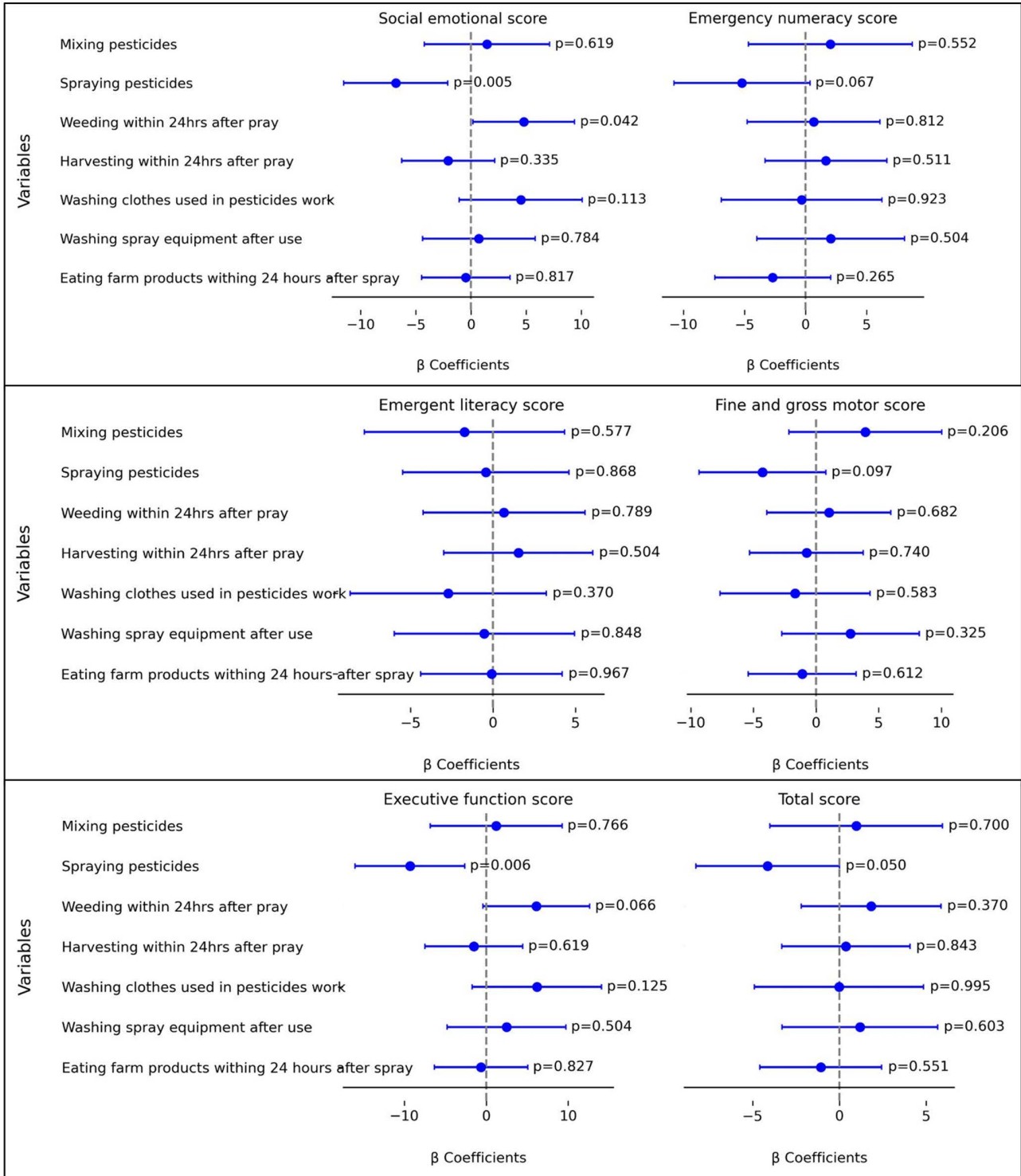

**Fig 2. The associations between 7 maternal self-reported pesticide exposure situations during pregnancy and their children's performance on the neurodevelopment tests IDELA domains at the of 4 to 6 years in Tanzania.** Adjusted β coefficients and 95% confidence interval are shown as results from multiple linear regression analysis, adjusted for the child's age.

aged between 7 and 12 years [32]. These findings in previous studies and the present one, suggest that early-life exposures to neurotoxic pesticides may contribute to impairments in core executive function domains such as short-term memory and inhibitory control.

There is also some inconsistence in the relationship between prenatal pesticide exposure and child neurodevelopment. For instance, in a Canadian cohort, first trimester maternal urinary glyphosate levels were not significantly associated with child neurodevelopment scores at 3–4 years of age [33]. Similarly, a study in Mexico City found that third-trimester levels of 3-PBA was not associated with changes in mental or motor development scores by 36 months [34]. In a Colombian cohort, where 13% of mothers reported spraying pesticides in the farm during pregnancy, the children had slightly lower but not statistically significant neurodevelopmental scores [35]. This evidence suggests that while some pesticide exposures show impact in children's early neurodevelopment, there are still inconsistencies among the studies which may be attributed to differences in exposure timing, type of pesticide studied, and how exposure and outcomes were measured. More research work is needed.

No significant associations were found in the present study between pesticide exposure from activities such as mixing pesticides, washing spray equipment, harvesting within 24 hours after pesticide spray or consuming crops within 24 hours of spraying. This may reflect differences in exposure intensity or variability in protective behaviors. For example, pesticide mixing may result in relatively lower exposure since it is less commonly performed by women, and when they do engage in it, they tend to use protective gear more consistently compared to when spraying [36]. Regarding pesticide ingestion, a quasi-experimental study of rural school children in Chile found that fruit consumption was associated with elevated concentrations of only one out of ten studied organophosphate metabolites [37]. This suggests that ingestion may not be a primary exposure route. However, findings from South Africa indicate that school children who consumed fruits directly from vineyards or orchards exhibited lower motor screening speed and visual processing accuracy scores [38].

Harvesting, a task predominantly performed by women [39], did not present a significant exposure risk in this study. In Kenya, individuals involved in harvesting on horticultural farms reported a higher frequency of pesticide-related symptoms [40]. Thus, pesticide exposure through harvesting exists and the study underlines that harvesting is mostly performed by women. In our study, children of mothers who engaged in harvesting had significantly lower fine and gross motor scores, though this association was not observed in the adjusted statistical models.

Children in this study population demonstrated higher neurodevelopmental scores (mean IDELA score: 51) compared to those from other low-resource setting countries. The overall mean IDELA scores in this study were slightly higher than those reported for children aged 3–6 years in Cape Town, South Africa (mean score: 48) [41]. The scores in the present study were clearly higher than among children aged 3–5 years, in a study from 29 villages in Cambodia (mean score: 43) [42]. This difference may be partly attributed to the age distribution of the study populations, which was 4–6 years in the current study, as all studies noted an upward trend in scores with increasing age. However, the scores in our present study were lower than those of preschool-aged children from low socioeconomic backgrounds in São Paulo, Brazil, who achieved an average IDELA score of 76.4 [43]. It is difficult to know why these differences are seen, but it might be related to different cultures and early education of the children [44].

This study demonstrates several methodological strengths; it is the first study to employ IDELA, a standardized and validated tool for objective evaluation of neurodevelopmental outcomes in children aged 3.5 to 6.5 years in low- and middle-income countries. By using IDELA, the study ensures precise and consistent assessments across critical developmental domains, including cognitive, motor and social emotional skills. Previous pesticide exposure studies used the Bayley Scales of Infant Development [45,46] and the Cambridge Automated NeuroPsychological Battery [38], yielding consistent findings. The children in our study had different age, and this may have influenced the results, as older children perform better than younger ones. This was controlled in the study by using statistical analyses to adjust for the age of the children. This adjustment was pivotal in ensuring that the observed relationships represent a more precise attribution of observed neurodevelopmental outcomes to pesticide exposure rather than extraneous factors.

However, the present study has several limitations. The reliance on self-reported data to assess pesticide exposure introduces potential recall bias, a common issue in similar research [47]. Such bias may compromise the validity of exposure estimates. Previous studies indicate that the accuracy of self-reported pesticide exposure depends on the timing and specificity of recall. When collected shortly after exposure or focused on general exposure information (e.g., hygiene practices), self-reports can yield relatively accurate predictions of outcomes like neurodevelopmental scores [48,49]. However, in this study, women were asked to recall farm activities from four to six years prior, which may have affected precision. For most women, the pesticide spraying is normally less frequent than other work tasks, and it is likely that they remember this better than other tasks, causing over-reporting of this type of work. No internal consistency checks between related exposure questions were conducted, as the different questions asked about work tasks and potential pesticide exposure were not expected to be associated. Also, the cross-sectional design poses inherent limitations, though these were mitigated through statistical methods examining associations between variables. While the study conducted multiple statistical tests, the potential for bias due to this was reduced by adopting a stricter significance threshold (p < 0.01).

In conclusion, our findings suggest that maternal involvement in pesticide-related activities, specifically pesticide spraying during pregnancy is associated with lower neurodevelopment scores, particularly in the social-emotional, executive function, and overall domains of their children. As the pesticide exposure is based on self-reports, these results should be interpreted with caution. Risk assessments, including pesticide exposure measurements, are clearly needed in horticulture, especially where the workers are women of reproductive age.

## Supporting information

**S1 Table. Linear regression models, with all independent variables included in the model simultaneously, were run to predict neurodevelopment scores from reported pesticide exposure activities using the child's age as a covariate, including both unadjusted and adjusted results.**
(XLSX)

**S2 Table. Sensitivity analyses, conducted to check whether the observed neurodevelopmental effects were sex-specific.** The analysis revealed that the significant negative association between spraying pesticides and the observed decrease in social-emotional and executive function scores is driven primarily by boys.
(XLSX)

**S1 File. Inclusivity-in-global-research-questionnaire 160925.**
(DOCX)

## Acknowledgments

We sincerely appreciate the support of the government authorities in the study areas during data collection. Most importantly, we are deeply grateful to all the women who participated in this study—their experiences and insights were invaluable in making this research possible.

## Author contributions

**Conceptualization:** William Nelson Mwakalasya, Simon Henry Mamuya, Bente Elisabeth Moen, Aiwerasia Vera Ngowi.

**Data curation:** William Nelson Mwakalasya, Simon Henry Mamuya, Karim Manji, Bente Elisabeth Moen, Aiwerasia Vera Ngowi.

**Formal analysis:** William Nelson Mwakalasya, Simon Henry Mamuya, Karim Manji, Bente Elisabeth Moen, Aiwerasia Vera Ngowi.

**Funding acquisition:** Simon Henry Mamuya, Bente Elisabeth Moen, Aiwerasia Vera Ngowi.

**Investigation:** William Nelson Mwakalasya, Simon Henry Mamuya, Karim Manji, Bente Elisabeth Moen, Aiwerasia Vera Ngowi.

**Methodology:** William Nelson Mwakalasya, Simon Henry Mamuya, Karim Manji, Bente Elisabeth Moen, Aiwerasia Vera Ngowi.

**Project administration:** Simon Henry Mamuya, Bente Elisabeth Moen, Aiwerasia Vera Ngowi.

**Resources:** William Nelson Mwakalasya, Simon Henry Mamuya, Karim Manji, Bente Elisabeth Moen, Aiwerasia Vera Ngowi.

**Supervision:** Simon Henry Mamuya, Bente Elisabeth Moen, Aiwerasia Vera Ngowi.

**Validation:** Aiwerasia Vera Ngowi.

**Writing – original draft:** William Nelson Mwakalasya.

**Writing – review & editing:** Simon Henry Mamuya, Karim Manji, Bente Elisabeth Moen, Aiwerasia Vera Ngowi.

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
