## [Decision Letter · Decision Letter 0]

4 Sep 2025

Dear Dr. Moen,

Thank you for submitting your manuscript to PLOS ONE. After careful consideration, we feel that it has merit but does not fully meet PLOS ONE’s publication criteria as it currently stands. Therefore, we invite you to submit a revised version of the manuscript that addresses the points raised during the review process.

We look forward to receiving your revised manuscript.

Kind regards,

Rajendra Prasad Parajuli, PhD

Academic Editor

PLOS ONE

Journal Requirements:

“This research was funded by the Norwegian Programme for Capacity Development in Higher Education and Research for Development (Norad). NORHED II – SAFEWORKERS Project, grant number: 69181”

Reviewers' comments:

Reviewer's Responses to Questions

**Comments to the Author**

1. Is the manuscript technically sound, and do the data support the conclusions?

Reviewer #1: No

Reviewer #2: Yes

2. Has the statistical analysis been performed appropriately and rigorously?

Reviewer #1: No

Reviewer #2: No

3. Have the authors made all data underlying the findings in their manuscript fully available?

Reviewer #1: No

Reviewer #2: No

4. Is the manuscript presented in an intelligible fashion and written in standard English?

Reviewer #1: Yes

Reviewer #2: No

Reviewer #1: Mwakalasya and colleagues investigated the impact of prenatal pesticide exposure during the first trimester on children's neurodevelopmental outcomes at ages 4 to 6. They employed a cross-sectional design with a sample of 432 mother–child dyads.

Exposure status during pregnancy was assessed retrospectively via maternal reports obtained when the children were 4 to 6 years old. Neurodevelopment was evaluated using a validated instrument, IDELA, which covers five neurodevelopmental domains.

The authors found that direct pesticide spraying by mothers during pregnancy was associated with child’s lower scores in certain neurodevelopmental domains, including socioemotional and executive functioning, but not in others. No other maternal exposure status was associated with child outcomes, except for weeding within 24 hours after spraying, which was unexceptionally associated with higher socioemotional scores.

Given the underrecognized risks of pesticide use in LMICs, the topic is highly relevant for public health policymakers and regulatory scientists. However, I will not prioritize this potential significance under the current PLOS ONE publication criteria, which focus on methodological rigor and clarity rather than public health urgency alone.

To assess whether the data support the conclusion, the manuscript requires further clarification or additional work on the following points.

1) The neurodevelopmental measures – needs clarification

The manuscript does not clearly describe how the IDELA domain scores were calculated, despite their importance as outcome variables. For instance, executive function (EF) domain comprises only 2 items, yet the average score for EF appers to hit 50 between age 5 to 6 as children grow. Is this T-score of average 50 with SD of 10? If not, how does a score of 50 arise from just two items? The current standard deviation seems much larger than expected for a T-score. Clarifying this is critical, as it will help readers interpret the magnitude of coefficients in the linear regression analyses shown later.

2) Uncertainty in exposure assessment – needs clarification and additional work

It appears that mothers were asked to recall exposure information from 4 to 6 years earlier. While it is plausible that pregnant women would remember potential pesticide exposures with high accuracy, the reliability of recall is only briefly acknowledged in the manuscript (line 271) - “women were asked to recall farm activities from four to six years prior, which may have affect precision (LL. 271)” - without further elaboration. The authors may strengthen this section by:

- Addressing whether any internal consistency checks (e.g., between responses to different questions) were conducted.

- Clarifying whether mothers who answered “No” to all exposure items (Table 2) had indeed not engaged in any horticultural activities during pregnancy.

3) Lack of specificity of the exposure status – needs clarification

While the authors should be commended for addressing pesticide risks even without specifying chemical types, the lack of specificity limits the manuscript’s contextualization in the broader literature. Numerous prior studies—including high-quality meta-analyses—have identified organophosphates, for example, as especially harmful to neurodevelopment. The authors could enhance the manuscript by:

- Providing contextual information about the types of pesticides most commonly used in Tanzania or the study area.

- Discussing how the unknown pesticide types might explain why only the socioemotional (but not literacy or numeracy) domains were affected.

4) Age effects on neurodevelopmental scores

Because IDELA scores increase with age, the authors appropriately included age as a covariate. However, more detail is needed:

- Was age entered in years or months?

- Why did the “Wasted” group perform better in all domains, particularly EF (Table 3)? Could this be due to older age in this group?

- The authors might consider reporting the number of children in each group in Table 3 to clarify these patterns.

A minor issue to be clarified/addressed.

LL. 144: What constitutes "delivery complications"? Definitions vary—some authors include twin births, others do not.

Reviewer #2: The study investigated the association between prenatal pesticide exposure and neurodevelopmental outcomes among children aged 4-6 years, using a cross-sectional survey data. The topic itself is important and timely, given growing concerns about pesticide exposures and child neurodevelopment. However, there are significant methodological concerns, particularly in statistical analyses, and several sections lack sufficient detail. My detailed comments are outlined below:

Introduction

Authors should minimize discussing why pesticides are used. Rater, they can discuss more about the health effect of pesticides and

The discussion on why pesticides are used could be minimized. Instead, the introduction should focus more on the health effects of pesticide exposure, as well as known biological mechanisms between exposure and the outcomes.

The rationale for conducting this study is poorly written. Especially, several cohort studies have previously investigated mental health and neurodevelopmental outcomes among children in-utero exposed to pesticides. The authors should clearly mention what gap this study fills and why this study (based on cross sectional design) would add a value to the literature.

Line 73: Authors mentioned that the present study focuses on pesticide exposure during the first trimester. However, first trimester exposure was not mentioned later in any other part.

Line 74-78: These sentences seem unnecessary.

Materials and Methods

Neurodevelopmental assessment: Authors need to explain more about the scales. Especially, how many items are included, how are items coded or scored; how the total IDELA score was calculated.

Please include a new sub-section on covariates. Please list out the covariates (including how they were measured/categorized), and clarify the rationale for including these covariates.

Statistical analysis: Authors note that the outcome variables (scores) are not normally distributed, however they used multiple linear regression to explore the relationship between exposure and the outcomes. Please clarify whether model assumptions were tested and met?

Also, I believe the IDELA scores are bounded within a fixed range (e.g., 0-50), they may represent censored continuous outcomes. In this case, Tobit models might be provides better results in this case.

Together with these scores as the (continuous) outcome variables, it might also be a good idea to categorize the outcome variables based on the cut-off scores, if any. Then other appropriate models could be used to explore the relationship between pesticide exposure and the categorical outcomes. In fact, it may offer robust insight and confirm associations.

Results:

Line 151: The manuscript interprets mean values in terms of percentages. Please revise the interpretations.

Figure 1: The title of the figure shows that the values are “mean neurodevelopmental score”. Usually, a boxplot present median and IQR values.

Line 179-187: The manuscript currently presents only adjusted results. Please present both unadjusted and adjusted beta coefficients. As mentioned above, the linear regression models might not be appropriate in this case.

Suggestions for additional analysis:

Authors may consider sensitivity analyses to confirm robustness of the findings.

Discussion:

Line 248-255: Percentage (%) are used with mean scores. Please clarify it for better readability.

**Do you want your identity to be public for this peer review?** For information about this choice, including consent withdrawal, please see our Privacy Policy

Reviewer #1: **Yes: ** Kenji J. Tsuchiya

Reviewer #2: No

---

## [Author Response · Author response to Decision Letter 1]

16 Sep 2025

Reviewer 1 Response

1) The neurodevelopmental measures – needs clarification

The manuscript does not clearly describe how the IDELA domain scores were calculated, despite their importance as outcome variables. For instance, executive function (EF) domain comprises only 2 items, yet the average score for EF appears to hit 50 between age 5 to 6 as children grow. Is this T-score of average 50 with SD of 10? If not, how does a score of 50 arise from just two items? The current standard deviation seems much larger than expected for a T-score. Clarifying this is critical, as it will help readers interpret the magnitude of coefficients in the linear regression analyses shown later.

Answer: We thank the reviewer for this crucial observation. We apologize for the lack of clarity. The IDELA tool does not use T-scores. The scoring is based on the percentage of tasks a child successfully completes within each domain. Each task is scored as 0 (incorrect/no response) or 1 (correct). The domain score is calculated as sum of points in the domain/Total possible points in the domain) * 100.

This results in a percentage score for each domain (range 0-100%). The Executive Function (EF) domain, for example, has a total of 17 possible points (short-term memory has 4 items, inhibitory control has 13 items) (Updated in Table 1). A score of 50% means the child correctly completed half of the tasks in that domain. We have added information to clarify this under "Neurodevelopmental assessment subsection (lines 124-129).

2) Uncertainty in exposure assessment – needs clarification and additional work

It appears that mothers were asked to recall exposure information from 4 to 6 years earlier. While it is plausible that pregnant women would remember potential pesticide exposures with high accuracy, the reliability of recall is only briefly acknowledged in the manuscript (line 271) - “women were asked to recall farm activities from four to six years prior, which may have affect precision (LL. 271)” - without further elaboration. The authors may strengthen this section by:

- Addressing whether any internal consistency checks (e.g., between responses to different questions) were conducted.

- Clarifying whether mothers who answered “No” to all exposure items (Table 2) had indeed not engaged in any horticultural activities during pregnancy.

Answer:

We agree with the reviewer that recall bias is an important limitation, and we have added text in the discussion part where this is mentioned:

“For most women, the pesticide spraying is normally less frequent than other work tasks, and it is likely that they remember this better than other tasks, causing over-reporting of this type of work. No internal consistency checks between related exposure questions were conducted, as the different questions asked about work tasks and potential pesticide exposure were not expected to be associated.” (lines 317-320)

The questions related to their work exposure were not analyzed for internal consistency, as they represent different types of work tasks. The questionnaire was designed to capture distinct exposure activities, and it is plausible for a participant to engage in one activity (e.g., weeding) but not another (e.g., mixing). We have added text in the introduction, lines 49-50 about this:

“These activities vary due to different needs in the farm. The women may have specific work tasks like weeding, but can be asked to perform other tasks for shorter or longer periods.”

Regarding mothers who answered “No” to all exposure items, we reviewed the data and found that there were no women who reported "No" to all 7 exposure activities.

3) Lack of specificity of the exposure status – needs clarification

While the authors should be commended for addressing pesticide risks even without specifying chemical types, the lack of specificity limits the manuscript’s contextualization in the broader literature. Numerous prior studies—including high-quality meta-analyses—have identified organophosphates, for example, as especially harmful to neurodevelopment. The authors could enhance the manuscript by:

- Providing contextual information about the types of pesticides most commonly used in Tanzania or the study area.

- Discussing how the unknown pesticide types might explain why only the socioemotional (but not literacy or numeracy) domains were affected.

Answer:

Thank you very much for this important observation. We have added the publication by Mrema et al., 2017 that recorded 43 pesticides used in horticulture in Tanzania. The study shows that majority of the pesticides were organophosphates (32.6%), synthetic pyrethroids (14%), dithiocarbamates (11.6%), carbamates (7%) and others. Therefore, we have added this information in the introduction: “There is a documented evidence of the use of some of these pesticides in horticulture in northern Tanzania” (lines 66-67) and discussion: “Though this study did not perform biological monitoring, there are evidences that some of the neurotoxic pesticides are in use by horticulture farmers in Tanzania” (lines 257-258) sections.

4) Age effects on neurodevelopmental scores

Because IDELA scores increase with age, the authors appropriately included age as a covariate. However, more detail is needed:

- Was age entered in years or months?

- Why did the “Wasted” group perform better in all domains, particularly EF (Table 3)? Could this be due to older age in this group?

- The authors might consider reporting the number of children in each group in Table 3 to clarify these patterns.

Answer:

We thank the reviewer for pointing out these issues. Age was entered into the regression models as a continuous variable in years. This has been clarified in the Covariates subsection (lines 137-141).

Regarding “wasted” children outperforming others, we re-examined the data. The mean age of children in the "wasted" group was 5.7 years was indeed slightly higher than in the "normal" children (5.4 years) and "overweight" (5.3 years) groups. This age difference likely explains their higher scores. We have added this interpretation to the results section when discussing Table 3 (lines 191-193).

The sample size for each category of the variables is stated both in table 3 and 4.

5. A minor issue to be clarified/addressed.

LL. 144: What constitutes "delivery complications"? Definitions vary—some authors include twin births, others do not.

Answer: Thank you for this observation. We have clarified this in the Materials and Methods section. Delivery complications in this study included cesarean delivery, obstructed labor, severe bleeding, sepsis and the like. It did not include uncomplicated twin births (lines 114-115).

Reviewer 2

Introduction

Authors should minimize discussing why pesticides are used. Rather, they can discuss more about the health effect of pesticides and the discussion on why pesticides use should be minimized. Instead, the introduction should focus more on the health effects of pesticide exposure, as well as known biological mechanisms between exposure and the outcomes.

The rationale for conducting this study is poorly written. Especially, several cohort studies have previously investigated mental health and neurodevelopmental outcomes among children in-utero exposed to pesticides. The authors should clearly mention what gap this study fills and why this study (based on cross sectional design) would add a value to the literature.

Line 73: Authors mentioned that the present study focuses on pesticide exposure during the first trimester. However, first trimester exposure was not mentioned later in any other part. Line 74-78: These sentences seem unnecessary.

Answer:

We thank the reviewer for this guidance. We have revised the introduction section by replacing some of the information discussing the rationale for pesticide use with the known health effects and biological mechanisms linking pesticide exposure to neurodevelopmental outcomes.

“Literature shows that pesticides exposure can cause severe acute and chronic health effects in humans, including neurotoxicity, carcinogenicity, endocrine disruption, and respiratory issues, with agricultural workers and communities in developing countries facing disproportionate exposure” (lines 54-56).

“Prenatal exposures are also linked with damage of the nervous system through several other non-cholinesterase mechanisms. These include impairing axonal transport, inducing oxidative stress, causing mitochondrial dysfunction, and triggering neuroinflammation.” (line 69-71).

The information from line 73 to 78 have been revised by improving the last paragraph (rationale) to be integrated more smoothly into the flow of the introduction (lines 80-86):

“Also, neurodevelopmental outcomes have been evaluated by diverse assessment tools across different age groups, but standardized tools have rarely been used [25] Also, children aged 4 to 6 years have rarely been studied, particularly not in sub-Saharan Africa. This study aims to investigate the relationship between self-reported maternal pesticide exposure during pregnancy and neurodevelopmental outcomes, assessed using the International Development and Early Learning Assessment (IDELA), in their children aged 4 to 6 years in Tanzania. The use of self-reported exposure is particularly relevant for designing practical risk communication and mitigation strategies in resource-limited settings where biomonitoring is often not feasible. “

Materials and Methods

Neurodevelopmental assessment: Authors need to explain more about the scales. Especially, how many items are included, how are items coded or scored; how the total IDELA score was calculated.

Please include a new sub-section on covariates. Please list out the covariates (including how they were measured/categorized), and clarify the rationale for including these covariates.

Statistical analysis: Authors note that the outcome variables (scores) are not normally distributed, however they used multiple linear regression to explore the relationship between exposure and the outcomes. Please clarify whether model assumptions were tested and met?

Also, I believe the IDELA scores are bounded within a fixed range (e.g., 0-50), they may represent censored continuous outcomes. In this case, Tobit models might be providing better results in this case.

Together with these scores as the (continuous) outcome variables, it might also be a good idea to categorize the outcome variables based on the cut-off scores, if any. Then other appropriate models could be used to explore the relationship between pesticide exposure and the categorical outcomes. In fact, it may offer robust insight and confirm associations.

Answer:

We thank the reviewer for pointing this out. We have improved Table 1 by adding a column with possible total score for each item. In addition, we have added information on how the IDELA scores were calculated (lines 124-129).

We have added a new subsection titled Covariates in the materials and methods section (lines 137-141).

Thank you very much for bringing this up. We acknowledge that linear regression may not be ideal for non-normal data. We chose it for its interpretability and because it is robust to violations of normality with a large sample size (n=432). We have now explicitly stated this justification and that we confirmed the residuals of the final models were approximately normally distributed. We also tested for homoscedasticity (lines 154-159).

The IDELA test scores have no censored values as the scores cannot go below 0. However, scores of 0 are not seen as a result during this testing, and Tobit models are not relevant.

Establishing the relationship between pesticide exposure and IDELA scores categories could have been a good idea. However, there are no pre-determined cut-off scores so far.

Results:

Line 151: The manuscript interprets mean values in terms of percentages. Please revise the interpretations.

Figure 1: The title of the figure shows that the values are “mean neurodevelopmental score”. Usually, a boxplot present median and IQR values.

Line 179-187: The manuscript currently presents only adjusted results. Please present both unadjusted and adjusted beta coefficients. As mentioned above, the linear regression models might not be appropriate in this case.

Suggestions for additional analysis:

Authors may consider sensitivity analyses to confirm robustness of the findings.

Answer:

Thank you very much for pointing this out. We have removed the percentage signs (check line 180 in the track changes document). However, every score data presented in the study is in a form of percentage. The mean presented here is the average of the percentage scores.

The figure title has been improved as suggested. Thank you (line 187).

Thank you. The unadjusted data were not presented because they could have resulted into a very long table. In response to your suggestion, we have added the data in the supplemental table (Table S1) and added a text that refers to the table in the results section (lines 222-223).

This is an important point, thank you for bringing it. We performed a sensitivity analysis to check whether the observed neurodevelopmental effects were sex-specific. The findings are summarized in Table S2 and the narration was added in the results section (lines 224-230).

Discussion:

Line 248-255: Percentage (%) are used with mean scores. Please clarify it for better readability.

Answer:

Thank you for pointing this out, the % in the discussion were removed as suggested (line 293-300).

---

## [Decision Letter · Decision Letter 1]

27 Oct 2025

Dear Dr. Moen,

Thank you for submitting your manuscript to PLOS ONE. After careful consideration, we feel that it has merit but does not fully meet PLOS ONE’s publication criteria as it currently stands. Therefore, we invite you to submit a revised version of the manuscript that addresses the points raised during the review process.

**Dear Dr. Bente Elisabeth Moen,**

Thank you for your revised submission titled “Neurodevelopment in children born to women exposed to pesticides during pregnancy” (Manuscript ID: PONE-D-25-27255R1).

We appreciate the improvements made to the manuscript. Based on the reviewers’ evaluations, we find your study to be of interest and value. However, a few minor issues remain that need to be addressed before we can proceed further.

We therefore invite you to revise the manuscript once more, carefully considering the reviewers' remaining suggestions and ensuring the clarity, consistency, and completeness of your responses and manuscript.

We look forward to receiving your revised submission.

We look forward to receiving your revised manuscript.

Kind regards,

Rajendra Prasad Parajuli, PhD

Academic Editor

PLOS ONE

Journal Requirements:

Additional Editor Comments:

Dear Dr. Bente Elisabeth Moen,

Thank you for your revised submission titled “Neurodevelopment in children born to women exposed to pesticides during pregnancy” (Manuscript ID: PONE-D-25-27255R1).

We appreciate the improvements made to the manuscript. Based on the reviewers’ evaluations, we find your study to be of interest and value. However, a few minor issues remain that need to be addressed before we can proceed further.

We therefore invite you to revise the manuscript once more, carefully considering the reviewers' remaining suggestions and ensuring the clarity, consistency, and completeness of your responses and manuscript.

We look forward to receiving your revised submission.

Reviewers' comments:

Reviewer's Responses to Questions

**Comments to the Author**

Reviewer #1: (No Response)

Reviewer #2: All comments have been addressed

2. Is the manuscript technically sound, and do the data support the conclusions?

Reviewer #1: Partly

Reviewer #2: Yes

3. Has the statistical analysis been performed appropriately and rigorously?

Reviewer #1: Yes

Reviewer #2: Yes

4. Have the authors made all data underlying the findings in their manuscript fully available?

Reviewer #1: Yes

Reviewer #2: Yes

5. Is the manuscript presented in an intelligible fashion and written in standard English?

Reviewer #1: Yes

Reviewer #2: Yes

Reviewer #1: I thank the authors to address all the comments.

Please specify - LL. 114-115: "and the like" needs further clarification.

A minor grammatical error in LL. 258: "evidence" is usually treated as uncountable/singular.

Also, LL. 67 and 259 - "some of the neurotoxic pesticides are in use" needs to be more specific; readers are entitled to understand which types of pesticides are regarded to as "some of them". Also the Mrema's finding would better be expanded with numerical information either in introduction or discussion.

Thank you again for your taking care of the revision.

Reviewer #2: (No Response)

**Do you want your identity to be public for this peer review?** For information about this choice, including consent withdrawal, please see our Privacy Policy

Reviewer #1: **Yes: ** Kenji J. Tsuchiya

Reviewer #2: No

---

## [Author Response · Author response to Decision Letter 2]

7 Nov 2025

Response to reviewer’s comments

We are grateful to the editors and the reviewers for their time and insightful comments on our manuscript titled “Neurodevelopment in children born to women exposed to pesticides during pregnancy”. Their feedback has been invaluable in strengthening our manuscript. We have carefully considered all points raised by reviewer 2 and have made revisions in the manuscript in response to the comments. Point-to-point response to the comments is summarized in the table below. The page numbers and lines refer to the manuscript with track changes.

Table 1: Response matrix to reviewer’s comments

Reviewer 1 Author’s response

All comments have been addressed. No additional comments We thank the reviewer after being satisfied with our response to the provided comments and for indicating that they have no further concerns following our revisions. We appreciate for the valuable contribution to improving our manuscript.

Reviewer 2

QPlease specify - LL. 114-115: "and the like" needs further clarification.

A: We appreciate the reviewer’s suggestion. The vague phrase “and the like” has been replaced with a clearer description of additional complications to enhance precision and readability. The revised sentence on page 5, line 100 reads:

“Delivery complications included conditions such as obstructed labor, severe bleeding and sepsis, as reported by the mother.”

QA minor grammatical error in LL. 258: "evidence" is usually treated as uncountable/singular.

A:We thank the reviewer for this valuable comment. The grammatical error has been corrected by replacing “there are evidences” with “there is evidence.” This is on page 17, line 244.

Q:Also, LL. 67 and 259 - "some of the neurotoxic pesticides are in use" needs to be more specific; readers are entitled to understand which types of pesticides are regarded to as "some of them". Also the Mrema's finding would better be expanded with numerical information either in introduction or discussion.

A:We thank the reviewer for this valuable comment. The sentence has been revised to specify the types of pesticides and to include quantitative information from reference 16 (Mrema et al). This is added on page 17, lines 244-248.

---

## [Editor Report · Decision Letter 2]

14 Nov 2025

Neurodevelopment in children born to women exposed to pesticides during pregnancy

PONE-D-25-27255R2

Dear Dr. Moen,

We’re pleased to inform you that your manuscript has been judged scientifically suitable for publication and will be formally accepted for publication once it meets all outstanding technical requirements.

Kind regards,

Rajendra Prasad Parajuli, PhD

Academic Editor

PLOS ONE

Additional Editor Comments (optional):

Before final acceptance, please make the following minor editorial adjustments:

Consolidate the tables so that they appear together on a single page, if possible.

The added sentence, “More than 80% of the analysed pesticide samples of this Tanzanian study contained pesticides classified as moderate to highly hazardous to health [16],” should appear only once (preferably in the Introduction) and not in bold in the Discussion.

If feasible, have the final version reviewed by a native English speaker to ensure clarity and flow.
---

## [Editor Report · Acceptance letter]

PONE-D-25-27255R2

PLOS One

Dear Dr. Moen,

I'm pleased to inform you that your manuscript has been deemed suitable for publication in PLOS One. Congratulations! Your manuscript is now being handed over to our production team.

Kind regards,

on behalf of

Dr. Rajendra Prasad Parajuli

Academic Editor

PLOS One